# Whole Exome Sequencing Reveals a Novel *AUTS2* In-Frame Deletion in a Boy with Global Developmental Delay, Absent Speech, Dysmorphic Features, and Cerebral Anomalies

**DOI:** 10.3390/genes12020229

**Published:** 2021-02-05

**Authors:** Pietro Palumbo, Ester Di Muro, Maria Accadia, Mario Benvenuto, Marilena Carmela Di Giacomo, Stefano Castellana, Tommaso Mazza, Marco Castori, Orazio Palumbo, Massimo Carella

**Affiliations:** 1Division of Medical Genetics, Fondazione IRCCS-Casa Sollievo della Sofferenza, 71013 San Giovanni Rotondo (Foggia), Italy; p.palumbo@operapadrepio.it (P.P.); e.dimuro@operapadrepio.it (E.D.M.); mariobenvenuto.foggia@gmail.com (M.B.); m.castori@operapadrepio.it (M.C.); m.carella@operapadrepio.it (M.C.); 2Medical Genetics Service, Hospital “Cardinale G. Panico”, 73039 Tricase (Lecce), Italy; m.accadia@piafondazionepanico.it; 3U.O.C di Anatomia Patologica, AOR Ospedale “San Carlo”, 85100 Potenza, Italy; marilena.digiacomo@ospedalesancarlo.it; 4Unit of Bioinformatics, Fondazione IRCCS Casa Sollievo della Sofferenza, 71013 San Giovanni Rotondo (Foggia), Italy; s.castellana@css-mendel.it (S.C.); t.mazza@css-mendel.it (T.M.)

**Keywords:** *AUTS2*, whole exome sequencing, neurodevelopmental disorders

## Abstract

Neurodevelopmental disorders (NDDs) are a group of highly prevalent, clinically and genetically heterogeneous pediatric disorders comprising, according to the Diagnostic and Statistical Manual of Mental Disorders 5th edition (DSM-V), intellectual disability, developmental delay, autism spectrum disorders, and other neurological and cognitive disorders manifesting in the developmental age. To date, more than 1000 genes have been implicated in the etiopathogenesis of NNDs. Among them, *AUTS2* (OMIM # 607270) encodes a protein involved in neural migration and neuritogenesis, and causes NNDs with different molecular mechanisms including copy number variations, single or multiple exonic deletion and single nucleotide variants. We describes a 9-year-old boy with global developmental delay, absent speech, minor craniofacial anomalies, hypoplasia of the cerebellar vermis and thinning of the corpus callosum, resulted carrier of the de novo *AUTS2* c.1603_1626del deletion at whole exome sequencing (WES) predicted to cause the loss of eight amino acids [p.(His535_Thr542del)]. Notably, our patient is the first reported so far in medical literature carrying an in-frame deletion and the first in which absent language, hypoplasia of the cerebellar vermis and thinning of the corpus callosum has been observed thus useful to expand the molecular spectrum of *AUTS2* pathogenic variants and to broaden our knowledge on the clinical phenotype associated.

## 1. Introduction

Neurodevelopmental disorders (NDDs) are a group of clinically and genetically heterogeneous disorders, which are diagnosed in childhood and encompass, but are not limited to, intellectual disability (ID), developmental delay (DD), autism spectrum disorders (ASDs), communication and learning disorders, attention deficit/hyperactivity disorders (ADHD) and developmental motor disorders. Emerging evidence is prompting to include epilepsy, developmental regression, sleep disturbance, mood and behavioral disorders, and aggression in the field of NDDs. Overall NNDs have an estimated prevalence of approximately 1–3% in the general population and represent one of the major challenges in medicine being the most frequent cause of disability in children [1]. The molecular milieu underpinning NDDs is increasingly complex with more than 1000 genes identified so far as implicated in their etiopathogenesis. Many of these genes converge on common pathways and protein networks, a fact that mirrors the clinical and molecular variability of NDDs.

Among them, genes involved in neuronal migration, extension, branching of the neurites, synaptic function, transcriptional regulation and construction of neuronal network are strongly represented [2]. *AUTS2* (OMIM #607270) belongs to a gene family involved in neural migration and neuritogenesis, pivotal steps to form a functional brain, and regulates these processes both at cytoplasmic and nuclear level. Molecular alterations of *AUTS2* causing NDDs currently include copy number variations (CNVs) and intragenic single or multi-exon deletions, as well as a restricted spectrum of single nucleotide variants, while patients carriers of likely pathogenic small in-frame variants involving functional relevant regions of the gene are still missing in medical literature.

In this report, we describe a 9-year-old boy with global developmental delay, absent speech, dysmorphic features, and cerebral anomalies, resulted carrier of a novel, de novo in-frame deletion of *AUTS2* (OMIM # 607270) detected at whole exome sequencing (WES).

## 2. Materials and Methods

### 2.1. Genomic DNA Extraction and Quantification

This family provided written informed consent to molecular testing and to the full content of this publication. This study was conducted in accordance with the 1984 Declaration of Helsinki and its subsequent revisions. Molecular testing carried out in this report is based on the routine clinical care of our institution. Peripheral blood samples were taken from both the proband and his parents, and genomic DNA was isolated by using Bio Robot EZ1 following manufacturer’s instructions (http://geneious.mx/catalogos/EZ1_AdvanXL_lr2.pdf (accessed on 1 February 2021)) (Quiagen, Solna, Sweden). The quality of DNA was tested on 1% electrophorese agarose gel, and the concentration was quantified by Nanodrop 2000 C spectrophotometer (Thermo Fisher Scientific, Waltham, MA, USA).

### 2.2. SNP-Array Analysis

High resolution SNP-array analysis on the proband’s DNA was carried out by using the CytoScan HD array (Thermo Fisher Scientific) as previously described [3]. Data analysis was performed using the Chromosome Analysis Suite Software version 4.1 (Thermo Fisher Scientific) following a standardized pipeline [4]. Briefly: (i) the raw data file (.CEL) was normalized using the default options; (ii) an unpaired analysis was performed using 270 HapMap samples as a baseline in order to obtain copy numbers value from .CEL files. The amplified and/or deleted regions were detected using a standard Hidden Markov Model (HMM) method. We retained CNVs ≥ 15 Kb in length and overlapping ≥ 10 consecutive probes to reduce the detection of false-positive calls. The significance of each CNV detected was determined by comparison with all chromosomal alterations identified in the patient to those collected in an internal database of ~4500 patients studied by SNP arrays since 2010, and public databases including Database of Genomic Variants (DGV; available on line at: http://projects.tcag.ca/variation/ (accessed on 1 February 2021)), DECIPHER (available on line at: https://decipher.sanger.ac.uk/ (accessed on 1 February 2021)), and ClinVar (available on line at: https://www.ncbi.nlm.nih.gov/clinvar/ (accessed on 1 February 2021)). Base pair positions, information about genomic regions and genes affected by CNVs, and known associated diseases have been derived from the University of California Santa Cruz (UCSC) Genome Browser (available online at: http://genome.ucsc.edu/cgi-bin/hgGateway (accessed on 1 February 2021)), build GRCh37 (hg19). The clinical significance of each rearrangement detected was assessed following the American College of Medical Genetics (ACMG) guidelines [5].

### 2.3. Whole Exome Sequencing (WES)

Proband’s DNA was analyzed by WES by using SureSelect Human All Exome V6 (Agilent Technologies, Santa Clara, CA, USA) following manufacturer instructions as previously described [6]. This is a combined shearing-free transposase-based library prep and target-enrichment solution, which enables comprehensive coverage of the entire exome. This system enables a specific mapping of reads to target deep coverage of protein-coding regions from RefSeq, GENCODE, CCDS, and UCSC Known Genes, with excellent overall exonic coverage and increased coverage of HGMD, OMIM, ClinVar, and ACMG targets. Sequencing was performed on a NextSeq 500 System (Illumina, San Diego, CA, USA) by using the Mid Output flow cells (300 cycles), with a minimum expected coverage depth of 100×. All variants obtained from WES were called by means of the HaplotypeCaller tool of GATK ver. 3.58 [7] and were annotated based on frequency, impact on the encoded protein, conservation, and expression using distinct tools, as appropriate (ANNOVAR, dbSNP, gnomAD, 1000 Genomes, EVS, ExAC, ESP, KAVIAR, and ClinVar) [8,9,10,11,12], and retrieving pre-computed pathogenicity predictions of ad-hoc tools from dbNSFP v 3.0 (e.g., PolyPhen-2, SIFT, Mutation Assessor, FATHMM, LRT and CADD) [13] and evolutionary conservation measures. Variants were discarded if reported as benign or likely-benign in ClinVar and/or if have a minor allele frequency (MAF) > 0.01. Next, variants prioritization was performed as following: (i) nonsense/frameshift variants in genes previously described as disease-causing by haploinsufficiency or loss-of-function; (ii) variants located in a critical or functional domain; (iii) variants affecting canonical splicing sites (i.e., ± 1 or ± 2 positions); (iv) variants absent in allele frequency population databases; (v) variant reported in allele frequency population databases, but with MAF significantly lower than expected for the gene; (vi) variant predicted and/or annotated as (probable) pathogenic in ClinVar and/or LOVD. Variant analysis was carried out considering the ethnicity of the proband.

Candidate variants were confirmed by Sanger sequencing in both the proband and the parents’ DNA. PCR products were sequenced by using BigDye Terminator v1.1 sequencing Kit following manufacturer’s instructions (Applied Biosystems, Foster City, CA, USA) and ABI Prism 3100 Genetic Analyzer (Thermo Fisher Scientific). The clinical significance of the identified putative variants was interpreted according to the American College of Medical Genetics and Genomics (ACMG) [14].

Nucleotide variants nomenclature follows the format indicated in the Human Genome Variation Society (HGVS, http://www.hgvs.org (accessed on 1 February 2021)) recommendations and reported in the Leiden Open Variation Databases (LOVD) (https://databases.lovd.nl/shared/variants/0000659804 (accessed on 1 February 2021)). The data have been deposited in the ArrayExpress database (https://www.ebi.ac.uk/arrayexpress/ (accessed on 1 February 2021)) under the accession number E-MTAB-10053.

Putative impact of candidate variants on protein was firstly assessed by searching for known functional/structural annotations in Uniprot [15]. Then, the reference FASTA sequence (Uniprot accession: Q8WXX7-1) was scanned for the presence of Eukaryotic Linear Motifs (ELM), using the ELM web-service (http://elm.eu.org/ (accessed on 1 February 2021)) [16]. Parameters were maintained at default values, while the “Taxonomy Context” was set to “Homo sapiens”. Results were filtered to consider only patterns with elevate conservation scores that overlapped with the mutant protein site.

## 3. Results

### 3.1. Clinical Description

This is a 9-year-old boy, second child of healthy non-consanguineous parents of Caucasian origin (southern Italy). No family history of congenital anomalies or ID/NDD was referred. He was born at 37 + 4 weeks of gestation by cesarean section for breech presentation. At birth, his weight was 3560 g (87th percentile), length 50 cm (66th percentile), head circumference was 34 cm (47th percentile), and Apgar scores were 9 and 10 at 1′ and 5′, respectively. The newborn was admitted to intensive care due to jaundice, limb hypertonia and facial dysmorphisms. Standard karyotype performed at birth was 46, XY. Subsequently, psychomotor development was severely delayed as he walked unsupported at 2.5 years and did not develop speech. At 7 years and 5 months, specialistic assessment showed cognitive delay. He did not acquire sphincter control and suffered from chronic constipation. He had difficulty falling asleep with partial improvement by melatonin intake. Parents reported only an episode of febrile seizure at 2 years old. Electroencephalogram at rest was normal. Brain MRI, showed hypoplasia of the cerebellar vermis and thinning of the corpus callosum. Echocardiogram and abdominal ultrasound were normal.

At last clinical evaluation, performed at 9 years old, he showed distinctive facial features including bitemporal narrowing, left posterior plagiocephaly, low anterior hairline, synophrys, strabismus, prominent nasal bridge, underdeveloped nasal alae, malar flattening, narrow palate, and slight anteverted ears (Figure 1). Examination of the oral cavity, limbs, extremities, skin and external genitalia was normal. Language was absent: he vocalized, but without communication purposes.

### 3.2. Molecular Findings

High-resolution SNP-array analysis did not identify any pathogenic copy number variations (microdeletions or microduplications) in the proband. The molecular karyotype of the patient, according with the International System For Human Cytogenetic Nomenclature (ISCN 2016), is: arr[GRCh37](1-22) × 2,(X,Y) × 1. WES revealed an in-frame deletion at heterozygous state in the exon 9 of the *AUTS2* gene (OMIM 607270) (*AUTS2*:NM_015570) c.1603_1626del resulting in a p.(His535_Thr542del) substitution (GRCh37/hg19). The variant was detected with a depth of coverage greater than 150×, and with elevate quality scores (i.e., Phread quality > 3000 and genotype quality = 99). This variation is not reported in gnomAD and ExAC populations’ database and it is predicted to cause the depletion of 4 out of 9 Histidine of one of the Histidine rich domain of the protein. Complete bioinformatics details are reported in Table 1.

The variant was confirmed by Sanger sequencing using specific primers (*AUTS2*, exon 9, Forward Primer: TGGTCTCGTCGTCTTCATTG; *AUTS2*, exon 9, Reverse Primer: CGTCAGTCCCCATTCGATCT). Parental DNA analysis showed that it is a de novo event (Figure 2).

The variant was classified as likely pathogenic according to ACMG guidelines [14]. No further variant classified as pathogenic or likely pathogenic, according to ACMG guidelines in other genes and previously associated with phenotypes compatible with the clinical features observed in the patients, were identified by the bioinformatics analysis.

This variant was predicted to cause the loss of a stretch of eight amino acids, including four histidines, [p.(His535_Thr542del)] in the histidine-rich domain 1 (amino acids 525-548) (Figure 3).

Within this compositionally biased region, the ELM resource predicted the presence of two high-scoring instances of the “MOD_GSK3_1” motif, at positions 531-538 and 535-542, respectively. These “HQHTHQHT” sequences would contain the linear motif *“...([ST])...[ST]”*, that in turn would represent the GSK3 phosphorylation site (http://elm.eu.org/elms/MOD_GSK3_1.html (accessed on 1 February 2021)). Along the whole AUTS2 sequence, four perfect matches were found for this motif. Motifs that perfectly match an ELM-annotated regular expression exhibit conservation scores of 1. Unfortunately, no crystal structures have been deposited (https://www.rcsb.org/ (accessed on 1 February 2021)) for the whole protein or protein fragments, while data on structural domains are scarce (details on https://www.uniprot.org/uniprot/Q8WXX7 (accessed on 1 February 2021)). However, the predicted linear motif co-locating the deletion would pave the way to functional studies.

## 4. Discussion

*AUTS2* comprises 19 exons, spans about 1.2 Mb of genomic DNA in the proximal 7q11.2 regions. Although the function of the gene has been poorly characterized for a long time, recent research papers provided strong evidence on its role in brain development. Interestingly, the AUTS2 protein has dual physiological roles: cytoplasmic AUTS2 regulates actin cytoskeleton to control neuronal migration and extension, while nuclear AUTS2 is involved in gene expression regulation of various genes [18,19,20,21]. Among the AUTS2 target genes identified, ~35.2% comprise the top 25% highly tran-scribed genes in mouse brain. Among them, *PRC1* and *SEMA5A*. Polycomb-group repressive complex 1 (*PRC1*), a polycomb-group gene involved in transcriptional repression, physically interacts with AUTS2, implicating a role for AUTS2 in developmental transcriptional regulation [22]. In 2013, the regulatory pathway for *SEMA5A* (semaphorin 5A), an autism candidate gene, was mapped in silico, using expression quantitative trait locus (eQTL) mapping. The authors found that the *SEMA5A* regulatory network significantly overlaps with rare CNVs around ASD-associated genes, including *AUTS2*. Performing eQTL mapping for expression levels of the eQTL-associated genes within the network (eQTLs of the eQTLs of *SEMA5A*), the authors identified 12 regions associated with the expression of 10 or more primary *SEMA5A* eQTL genes, including *AUTS2*. This study suggests that *AUTS2* is involved, and may be a master regulator in ASD-related pathways [23]. Sequence analysis of AUTS2 identified two proline-rich domains (amino acids 288-471 and 545-646), two histidine-rich domains (HR1 and HR2) (amino acids 525-548 and 1122-1181) and a predicted PY motif (PPPY) (amino acids 515-519) (domains available at https://www.uniprot.org/uniprot/Q8WXX7#family_and_domains (accessed on 1 February 2021)) [17]. The PY motif is a potential WW-domain-binding region involved in protein-protein interactions. Moreover, it is present in the activation domain of several transcription factors, suggesting the involvement of *AUTS2* in transcriptional regulation. Other predicted protein motifs include several cAMP and cGMP-dependent protein kinase phosphorylation sites and putative N-glycosylation sites. Functional evidence and data from the literature demonstrate that the expression of *AUTS2* is regulated by a well characterized post-mitotic projection-neuron specific transcription factor, TBR1, which binds the *AUTS2* promoter and activates the gene in developing neocortex [24].

*AUTS2* was firstly linked to a clinical condition in a paper by Sultana et al., in which the gene was found disrupted due to a balanced translocation in a pair of monozygotic (MZ) twins with ASD. In addition, the authors observed that *AUTS2* was strongly expressed in human fetal brain (frontal, parietal and temporal regions, telencephalon, ganglionic eminence, cerebellum anlagen, medulla oblongata, cortical plate and ventricular zone) with high expression in regions associated with higher cognitive brain functions [17]. Since then, more than 50 unrelated patients with neurodevelopmental (ID, ASD, speech and language disorders) and/or neuropsychiatric disorders (schizophrenia, ADHD, dyslexia and depression as well as addiction-related behaviors) have been shown to carry distinct heterozygous alterations of the *AUTS2* gene [25,26,27,28,29,30,31,32,33,34,35,36,37,38,39,40,41,42]. For these reasons, the term “*AUTS2* syndrome” was coined to describe the wide spectrum of phenotypes, predominantly affected the cognition, observed in individuals with a germline *AUTS2* alteration [28]. In addition to a variable NDD, satellite features frequently observed in affected individuals are feeding difficulties, short stature, hypotonia, and cerebral palsy, minor craniofacial anomalies.

From a clinical perspective, our patient shows several clinical manifestations associated to “*AUTS2* syndrome” namely growth problems, dysmorphic features, skeletal abnormalities. In addition, he showed absent language at 9 years of age, and hypoplasia of the cerebellar vermis and thinning of the corpus callosum, which have never been reported in this condition. Thus, the description of our patient is useful to better delineate the clinical phenotype associated to the syndrome. Obviously, further reports will help in evaluating the significance of such a provisionally novel association.

From a molecular perspective, to our knowledge, this is the first reported individual with a small in-frame deletion in *AUTS2*, which involve 4 out of 9 Histidine of the first Histidine-rich domain of the protein. This variant, which results to be de novo, is absent from population databases such as TOPMED and gnomAD, thus useful to expand the mutational spectrum of *AUTS2*. A putative functional role of the motif involved by this in-frame deletion might be theorized/hypothesized by considering the computational screening of ELM along the AUTS2 amino acid sequence. Short linear motifs are one of the main components of proteins, consisting of functional modules, usually with a length of 3–10 amino acids. Among the functions that they mediate there are: protein-protein interactions, targeting signals, degradation signals, phosphorylation sites or affinity control. Abnormalities of ELMs by genetic alterations or alternative splicing can provide different isoforms of some proteins with different or altered functionality. The “HQHTHQHT” sequences deleted in this subject would contain a linear motif that in turn seems to represent the GSK3 phosphorylation site. GSK3, a serine/threonine protein kinase, comprises two highly related proteins (GSK3-α and GSK3-β) that phosphorylate a wide variety of target proteins with a final inhibitory effect.

In our case, the deletion of part of the HR1 motive could have caused the loss of a phosphorylation site which could be the molecular etiopathogenetic mechanism underlying the clinical phenotype observed in the patient. In supported by additional observations and functional studies, our finding could shed more lights on the molecular pathogenesis of *AUTS2* syndrome.

Furthermore, our case suggest to further investigate the biological role of the His-rich motives, in order to elucidate their role in protein function and regulation. Obviously, being this the first patient carrying this kind of genetic variation of *AUTS2*, further functional studies are needed to confirm the pathogenic mechanism supposed.

## 5. Conclusions

The subject presented here is the first known individual with NDD and carrying an in-frame deletion of *AUTS2*. Our findings provisionally expand the *AUTS2* syndrome associated clinical spectrum to absent speech, hypoplasia of the cerebellar vermis and hypoplasia of the corpus callosum. Taken together, these findings expand the mutation spectrum of *AUTS2* syndrome and pave the way to a deeper understanding if its molecular pathogenesis.

## Figures and Tables

**Figure 1 genes-12-00229-f001:**
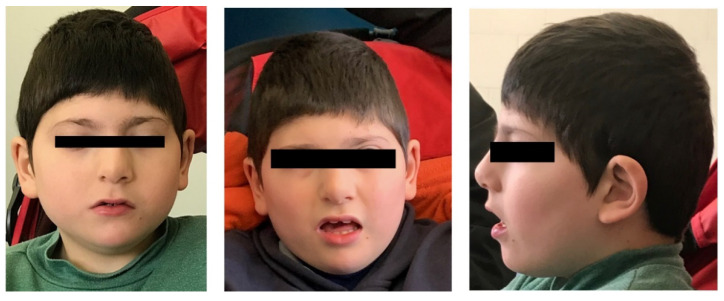
Craniofacial features observed in the investigated subject.

**Figure 2 genes-12-00229-f002:**
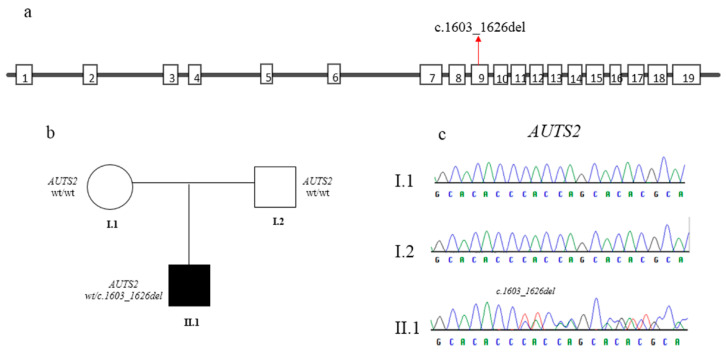
(**a**) Schematic representation of the structure of *AUTS2* gene. The variant identified here is indicated by red arrow. (**b**) Pedigree of the family displaying the de novo onset of the variant. Filled and unfilled circles/squares represent affected and unaffected individuals respectively. (**c**) Electropherograms of the proband (II.1) and his parents (I.1, I.2).

**Figure 3 genes-12-00229-f003:**
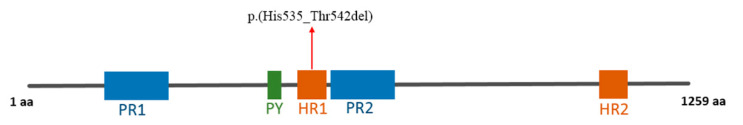
Schematic representation of the structure of AUTS2 protein [17]. The variant identified here is indicated by a red arrow (PR: proline-rich domains; PY: py domain; HR: histidine-rich domains).

**Table 1 genes-12-00229-t001:** Characteristics of the variant identified in the *AUTS2* gene.

Chromosome	Start	End	Reference Allele	Alternative Allele	Genotype	Gene	Nucleotide Change	Amino Acid Change	dbSNP ID	gnomAD_exome Allele Count	TOPMED Allele Count	ExAC_ALL Allele Count
**7**	70231220	70231244	AGCACCAGCACACCCACCAGCACAC	A	Heterozygous	*AUTS2* *NM_015570*	c.1603_1626del	p.(His535_Thr542del)	N.A.	N.A.	N.A.	N.A.

## Data Availability

The data presented in this study are openly available in ArrayExpress database (https://www.ebi.ac.uk/arrayexpress/) under the accession number E-MTAB-10053.

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
