# Peer review of "Whole Exome Sequencing Reveals a Novel AUTS2 In-Frame Deletion in a Boy with Global Developmental Delay, Absent Speech, Dysmorphic Features, and Cerebral Anomalies"

_genes, 2021, doi:10.3390/genes12020229_

Round 1

Reviewer 1 Report

The current manuscript titled ‘Novel AUTS2 in-frame deletion in a boy with global developmental delay, absent speech, dysmorphic features, and cerebral anomalies’ by Pietro Palumbo et al., is well written. However, the following points need to be addresses.

1# The title is not clear. They need to rearrange it.

2# In the Materials and Methods section, the authors need to cite reference from which they adopt/follow the genomic DNA extraction procedure. Furthermore, they need to follow this suggestion throughout the manuscript. They need to cite references for every procedure they followed.

3# In page 4 line 170-173, the authors wrote that ‘The variant was confirmed by Sanger sequencing using specific primers (AUTS2, 170 exon 9, Forward Primer: TGGTCTCGTCGTCTTCATTG; AUTS2, exon 9, Reverse Primer: CGTCAGTCCCCATTCGATCT).’ However, they did not display the result. Need to provide the figure of validation with positive control.

4# In page 5 line 183-185, the authors wrote that ‘This variant was predicted to cause the loss of a stretch of eight amino acids, including four histidines, [p.(His535_Thr542del)] in the histidine-rich domain 1 (amino acids 525-548)’. Does this variation cause any physical change in AUTS2 protein structure? Does the biological function of the AUTS2 protein change? The authors need to clarify it and provide an in depth description.

5# In the discussion, the authors wrote ‘nuclear AUTS2 is involved in gene expression regulation of various genes’. However, they did not mention the candidate target gene names in their manuscript. They need to summarize the target genes and choose some genes to show expressional differences.

6# In the discussion, the authors wrote ‘We report a boy with global developmental delay, absent speech, dysmorphic features and cerebral anomalies, in which a novel, de novo in-frame deletion of AUTS2 was detected by WES.’ This sentence is misleading. It seems to me that the in-frame deletion of AUTS2 is responsible for those phenotypic abnormalities. However, the authors did not showed any evidences in their manuscript.

7# They need to submit the Whole Exome Sequencing (WES) data to any of the publicly available platforms before a final decision of this current manuscript.

8# The sample size of this manuscript is only one. I would like to suggest claiming it as a case report rather than a full-length article.

Reviewer 2 Report

The manuscript entitled: “Novel AUTS2 in-frame deletion in a boy with global develop- 2 mental delay, absent speech, dysmorphic features, and cerebral 3 anomalies”  by Palumbo et al.  Manuscript ID: genes-1090423) describes a new molecular basis of the neurodevelopmental disorder in a 9-year-old boy.

The following point should be addressed:

  1. Are there any limitation(s) of the study?
